# COF-Based Photocatalysts for Enhanced Synthesis of Hydrogen Peroxide

**DOI:** 10.3390/polym16050659

**Published:** 2024-02-29

**Authors:** Deming Tan, Xuelin Fan

**Affiliations:** 1School of Mechanical Engineering, Chengdu University, Chengdu 610106, China; 2Institute for Advanced Study, Chengdu University, Chengdu 610106, China

**Keywords:** covalent organic framework, H_2_O_2_ photosynthesis, photosynthesis, solar conversion

## Abstract

Covalent Organic Frameworks (COFs), with their intrinsic structural regularity and modifiable chemical functionality, have burgeoned as a pivotal material in the realm of photocatalytic hydrogen peroxide (H_2_O_2_) synthesis. This article reviews the recent advancements and multifaceted approaches employed in using the unique properties of COFs for high-efficient photocatalytic H_2_O_2_ production. We first introduced COFs and their advantages in the photocatalytic synthesis of H_2_O_2_. Subsequently, we spotlight the principles and evaluation of photocatalytic H_2_O_2_ generation, followed by various strategies for the incorporation of active sites aiming to optimize the separation and transfer of photoinduced charge carriers. Finally, we explore the challenges and future prospects, emphasizing the necessity for a deeper mechanistic understanding and the development of scalable and economically viable COF-based photocatalysts for sustainable H_2_O_2_ production.

## 1. Introduction

The quest for sustainable and clean energy sources has become a paramount concern in the face of escalating global energy demands and the pressing challenges of climate change. Solar energy, being abundant and renewable, stands out as a promising candidate to address these challenges. Efficiently harnessing the vast potential of solar energy not only offers a solution to the impending energy crisis but also holds the promise of reducing the carbon footprint associated with conventional fossil fuels. Photocatalysis is a process that uses light to drive chemical reactions and has emerged as an important technology [1,2,3,4,5,6]. By converting solar energy into chemical energy, photocatalysis offers a dual advantage [4,7]: it provides an avenue for sustainable energy storage and paves the way for the synthesis of valuable chemicals, including H_2_, CO, and hydrogen peroxide (H_2_O_2_).

Among various chemicals synthesized by photosynthesis, H_2_O_2_ has attracted significant attention [2,8,9,10]. H_2_O_2_ is valuable for its multifaceted applications and environmentally amicable nature, and has become an indispensable chemical field like industrial processes, environmental remediation, and healthcare and medical applications [11,12,13,14]. Its ability to decompose into water and oxygen underpins its appeal as an eco-friendly oxidant, minimizing the risk of generating secondary pollutants. The conventional anthraquinone oxidation process [15], which has been the industrial applicable for H_2_O_2_ production, is increasingly being scrutinized for its inherent drawbacks. These include not only the substantial energy consumption and the utilization of hazardous substrates but also the generation of significant amounts of waste, which poses considerable environmental and economic challenges. In light of these limitations, the researchers have pivoted towards seeking alternative, sustainable, and cleaner methods for H_2_O_2_ production, with a particular emphasis on minimizing environmental repercussions. Photocatalytic synthesis of H_2_O_2_ has emerged as a compelling alternative, offering the prospect of harnessing solar energy directly to drive chemical reactions, thereby circumventing the need for energy-intensive processes and deleterious chemicals [16,17]. This approach not only aligns with the global shift towards sustainable energy but also presents a pathway for the localized, on-demand production of H_2_O_2_, reducing the need for storage and transportation.

In the realm of photocatalytic H_2_O_2_ synthesis, the role of catalysts is paramount, dictating the efficiency, selectivity, and stability of the photocatalytic processes. Conventional photocatalysts are predominantly based on precious metals such as platinum [18], palladium [19], and gold [20,21], demonstrating commendable performance in facilitating the production of H_2_O_2_. However, the deployment of these noble metal-based catalysts is significantly hampered by their scarcity in the Earth’s crust, which intrinsically leads to high costs and poses sustainability concerns, especially in the context of large-scale applications and global accessibility. Consequently, the exploration of alternative non-metal-based photocatalysts has become a focal point in contemporary research, aiming to circumvent the limitations associated with noble-metal catalysts. Metal-free photocatalysts, particularly those based on linear polymers [22], polymeric carbon nitride (PCN) [23,24], polymer resins [25], supramolecular coordination [26,27], and covalent organic frameworks (COFs) [2,28,29], have emerged as promising candidates, offering the advantages of abundance and low cost under photocatalytic conditions. Among these, COFs, with their intrinsic porosity, tunable structures, and the ability to incorporate a myriad of organic functional groups (Figure 1), have garnered substantial attention since 2020 [30,31,32].

COFs are typically a class of porous polymers formed by organic building blocks connected through covalent bonds. They were first synthesized by Yaghi et al. under solvothermal conditions through the self-condensation of phenyl diboronic acid (PDBA) and the co-condensation of PDBA with hexahydroxytriphenylene (HHTP) [33]. This work opened the door to COF research. Subsequently, various methods for synthesizing COFs have been reported, including solvothermal [34,35,36,37,38], microwave [39,40,41], ionothermal [42,43,44], and mechanochemical methods [45,46] for powder synthesis, and interfacial methods [47,48,49] for thin-film synthesis. At the same time, a wide variety of organic building blocks and linkages have been reported. To date, reported linkages include boroxine [50], boronate-ester [51,52,53], imine [54,55,56,57], hydrazone [58,59], squaraine [60,61,62], azine [63,64,65], imide [66,67], C=C [68,69,70,71], 1,4-dioxin linkage [72,73], among others. The COFs synthesized by these methods have shown great potential in applications such as sensing [74,75,76], catalysis [5,77,78,79], energy storage and conversion, [6,80,81,82,83] organic electronic devices [84,85,86,87,88,89,90,91,92], etc. [6,80,81,82,83,93,94,95,96].

COFs not only possess ordered pore structure and high surface area but also offer the possibility of fine-tailoring their electronic and structural properties to optimize charge separation and transfer, thereby standing out as a promising class of materials in the ongoing quest for sustainable and efficient photocatalytic processes. Despite the growing number of literature reports on COF photocatalytic synthesis of H_2_O_2_ since 2020, there is a scarcity of review papers summarizing the principal advancements in this field. Therefore, this review aims to acquaint readers with the advancement in this nascent field, as well as the prevailing challenges. Finally, perspectives are provided on the future development of this field.

## 2. Principles of Photocatalytic H_2_O_2_ Generation

Equation (1) illustrates the full process of photocatalytic synthesis of H_2_O_2_. This procedure encompasses two distinct half-reactions, namely the oxygen reduction reaction (ORR) (Equations (2) and (3)) and the water oxidation reaction (WOR) (Equation (4)). In response to the impetus of photons, the electrons within the catalyst undergo a transition from the valence band (VB) to the conduction band (CB), thus giving rise to the emergence of photo-excited *h^+^* and *e*^−^ species. The subsequent migration of these charges to the catalyst’s surface facilitates their active participation in a cascading series of reduction-oxidation reactions (Figure 2a), thereby selectively yielding H_2_O_2_.

The photo-excited *e*^−^ with reducibility is capable of reacting with oxygen through a 2*e*^−^ ORR process for the generation of H_2_O_2_ (see Equation (2)). It is worth noting that the oxygen involved in this reaction may be derived from either the 4*e*^−^ WOR or the atmospheric oxygen. Theoretically, as shown in Figure 2b, the 2*e*^−^ ORR can proceed directly through a 2*e*^−^ process (O_2_ + 2*e*^−^ + 2H^+^ → H_2_O_2_) or two consecutive 1*e*^−^ reactions via the superoxide radical intermediate (·O_2_^−^) (III of Figure 2b). In parallel, the photo-excited *h^+^*, which exhibits oxidizability, enables water molecules to generate H_2_O_2_ through a directly one-step 2*e*^−^ WOR (see Equation (3)) or two-step 1*e*^−^ WOR, or to produce O_2_ through a 4*e*^−^ WOR (see Equation (4)). Similar to the two-step 1*e*^−^ ORR process, in the two-step 1*e*^−^ WOR process, the photo-induced proton *h^+^* can initially oxidize H_2_O to generate a hydroxyl radical (·OH) intermediate. Subsequently, H_2_O_2_ is formed indirectly through the combination of two ·OH (VI of Figure 2b). These intermediates can be identified through in-situ characterization techniques. In-situ diffuse reflectance infrared Fourier transform spectroscopy is extensively utilized for the characterization of intermediates in the photocatalytic generation of H_2_O_2_. A well-designed photocatalyst aimed at the overall reaction should possess functional groups that effectively promote both the 2*e*^−^ ORR and 2*e*^−^ WOR. These two reactions, in turn, serve to maintain charge balance by, respectively, consuming *e*^−^ and *h^+^*. In instances where catalysts are specifically engaged in one of the half-reactions, sacrificial reagents are often employed to consume uninvolved charges, thereby safeguarding the catalyst and directing the reaction’s selectivity.
(1)$${2H}_{2}O+O_{2}\overset{hv}{\to }2H_{2}O_{2}$$
(2)$$O_{2}+2e^{-}+2H^{+}\to H_{2}O_{2}$$
(3)$${2H}_{2}O+2h^{+}\to H_{2}O_{2}+2H^{+}$$
(4)$$H_{2}O+4h^{+}\to O_{2}+4H^{+}$$

Nonetheless, it is worth noting that the solar-to-chemical energy conversion efficiency (SCC) in the context of photocatalytic H_2_O_2_ synthesis remains relatively low at present, seldom surpassing 1% [97]. This efficiency discrepancy is quite pronounced when compared to the performance observed in photocatalytic hydrogen production from water [98,99]. Several factors contribute to this relatively low efficiency, including limited light absorption, a propensity for charge recombination, and challenges in achieving desirable selectivity towards H_2_O_2_ [97,100,101]. Particularly, achieving selectivity towards H_2_O_2_ presents a significant hurdle (Figure 2). The WOR process commonly tends to favor O_2_ production via the 4*e*^−^ pathway rather than H_2_O_2_ production through the 2*e*^−^ pathway [17]. Additionally, side reactions and H_2_O_2_ decomposition also impact both the yield and selectivity [102,103]. Through the desirable optimization of catalyst structures and reaction conditions, significant enhancements can be achieved in terms of the yield and selectivity of photocatalytic H_2_O_2_ synthesis.

## 3. Evaluation of Photocatalytic H_2_O_2_ Generation

The evaluation criteria for photocatalytic H_2_O_2_ production mainly include H_2_O_2_ production rate, apparent quantum yield (AQY), and solar-to-chemical energy conversion (SCC) efficiency. The rate of H_2_O_2_ production is a critical parameter for quantifying the speed of H_2_O_2_ formation in photocatalytic processes. This rate is commonly delineated in millimoles per hour per gram (mmol h^−1^ g^−1^), referencing the catalyst’s specific mass. AQY quantifies the efficiency of H_2_O_2_ generation in relation to photon absorption by the catalyst, which can be calculated by Formula (5). So far, the highest reported AQY value for a COF catalyst is 38% [10]. SCC measures the efficiency of transforming solar energy into chemical energy stored in H_2_O_2_ and can be calculated by Formula (6), in which the total input power (W) can be obtained by multiplying the irradiation intensity of the Xenon lamp by the irradiated area and time.
(5)$$AQY\left(\%\right)=\frac{\left[H_{2}O_{2}\mathrm{produced}\left(\mathrm{mol}\right)\right]\times 2}{\mathrm{photon}\mathrm{number}\mathrm{entered}\mathrm{into}\mathrm{the}\mathrm{reactor}\left(\mathrm{mol}\right)}\times 100\%=\frac{[N_{a}\times h\times c]\left[H_{2}O_{2}\mathrm{produced}\left(\mathrm{mol}\right)\right]\times 2}{I\times S\times t\times \lambda }\times 100\%$$
where,
*N*_a_ (Avogadro’s constant) = 6.02 × 10^23^ mol^−1^;*h* (Planck constant) = 6.626 × 10^−34^ J·s;*c* (Light speed) = 3 × 10^8^ m/s;S = Irradiation area (cm^2^) = 11.5 cm^2^;I = The intensity of irradiation light (W/cm^2^);t = The photoreaction time (s) = 3600 s;λ = The wavelength of the monochromatic light (nm) = 420 × 10^−9^.
(6)$$\mathrm{SCC}\mathrm{effeciency}\left(\%\right)=\frac{\left[\Delta G\mathrm{for}H_{2}O_{2}\mathrm{generation}\left({J\mathrm{mol}}^{-1}\right)\right]\left[H_{2}O_{2}\mathrm{formed}\left(\mathrm{mol}\right)\right]}{\left[\mathrm{Total}\mathrm{input}\mathrm{power}\left(W\right)\right]\left[\mathrm{Reaction}\mathrm{time}\left(s\right)\right]}\times 100\%$$where ΔG is 117 kJ mol^−1^.

## 4. COFs for Light-Driven H_2_O_2_ Production

In the realm of photocatalytic H_2_O_2_ synthesis, COF photocatalysts have demonstrated significant potential for application. These photocatalysts can be primarily categorized into three main types: firstly, those dedicated to catalyzing the 2*e*^−^ ORR; secondly, those focusing on the 2*e*^−^ WOR; and thirdly, those capable of simultaneously catalyzing both 2*e*^−^ ORR and 2*e*^−^ WOR. Compared to the 2*e*^−^ WOR process, the 2*e*^−^ ORR is relatively easier to achieve. As a result, most of the current research on COF-based photocatalytic production of H_2_O_2_ is realized through optimizing the 2*e*^−^ ORR process.

To further enhance the catalytic efficiency of the 2*e*^−^ ORR and WOR, researchers often attempt to introduce specific active sites to facilitate the reactions. According to literature, the discovered active sites for the 2*e*^−^ ORR include pyridazine [104], pyridine [105], pyrene [28], bipyridine [106], benzene [107,108], and diarylamine [109], while those for 2*e*^−^ WOR comprise bipyridine [106], triazine [110], diacetylene [111,112], acetylene [111,112], among others. The incorporation of these active sites not only improves the efficiency of COF photocatalysts in H_2_O_2_ synthesis but also provides valuable insights for a deeper understanding and optimization of these two crucial reaction processes. From Table 1, it can be observed that the AQY mostly ranges between 1% and 16%, the H_2_O_2_ production rate primarily ranges from 100 to 7500 μmol·h^−1^·g^−1^, and the SCC is relatively low, with the highest being 0.82%. Through in-depth studies and optimization of active sites, it is anticipated that COF photocatalysts will achieve better performance in the field of photocatalytic H_2_O_2_ synthesis. In the following section, we will elaborate in detail on the strategies for introducing active sites.

### 4.1. Functional Groups Modification

Functional group modification in COF photocatalysts is a pivotal strategy that involves the intricate alteration or introduction of specific chemical groups within the COF structure to enhance their photocatalytic efficiency. Firstly, the tailored functional groups can optimize the electronic properties of COFs, enhancing visible light absorption and improving the charge separation efficiency. Additionally, the modified functional groups can enhance the stability of COFs under various operational conditions, ensuring consistent performance over extended periods. Furthermore, the strategic incorporation or withdrawing groups aids in the efficient transfer of charges, reducing the recombination of electron-hole pairs and amplifying the yield of H_2_O_2_. Luo et al. explore the use of sulfone-modified COFs as photocatalysts for H_2_O_2_ generation (Figure 3a) [113]. Sulfone units were introduced into the COFs as a strong electron-accepting core. Under LED visible light irradiation, FS-COFs exhibited an H_2_O_2_ yield of 1501.6 μmol·h^−1^·g^−1^ from pure water (Figure 3b), which was approximately three times more efficient than COFs with a similar structure but without sulfone sites (yield of 487.6 μmol·h^−1^·g^−1^). Notably, under Xe light irradiation, FS-COFs achieved an H_2_O_2_ yield of 3904.2 μmol·h^−1^·g^−1^. Mechanism study reveals that the incorporation of sulfone units into COFs facilitated the direct reduction in O_2_ to H_2_O_2_ via the one-step 2*e*^−^ ORR route. This was attributed to the improved separation of *e*^−^-*h^+^* pairs, enhanced protonation property, and altered O_2_ adsorption configuration on FS-COFs. The formation of 1,4-endoperoxide intermediate species favored the direct reduction in O_2_ to H_2_O_2_, avoiding the generation of reactive radicals and promoting higher H_2_O_2_ yields.

Similarly, Sun et al. introduced pyrene into COFs and obtained a series of pyrene-based COFs [28]. The performance of pyrene-based COFs was compared with other COFs, and the results showed that the presence of pyrene active sites in a dense concentration in a limited surface area led to undesirable H_2_O_2_ decomposition. This finding indicated that the distribution of pyrene units over a large surface area of the COFs played a crucial role in catalytic performance. To address the issue of H_2_O_2_ decomposition, a two-phase reaction system (water-benzyl alcohol) was employed. This approach effectively inhibited H_2_O_2_ decomposition and allowed for efficient photocatalytic H_2_O_2_ generation. Density Functional Theory (DFT) calculations revealed that the pyrene unit was more active for H_2_O_2_ production compared to bipyridine and (diarylamino)benzene units. In the presence of the sacrificial agent benzyl alcohol and under illumination with a light source ranging from 420–700 nm, the photocatalytic yield of H_2_O_2_ was 1242 μmol·h^−1^·g^−1^, with an AQY of 4.5% at 420 nm. In the absence of a sacrificial agent, satisfactory catalytic performance can still be achieved in H_2_O_2_ synthesis. Wu and co-workers synthesized thiourea-functionalized Covalent Triazine Framework (Bpt-CTF) (Figure 4a), which demonstrates a photocatalytic H_2_O_2_ production rate of 3268.1 μmol·h^−1^·g^−1^ (Figure 4b,c) without the necessity for sacrificial agents or cocatalysts [114]. This represents an improvement of over an order of magnitude compared to the unfunctionalized CTF (Dc-CTF), and it exhibits an AQY of 8.6% at 400 nm. Mechanistic studies demonstrate that the introduction of polar (thio)urea moieties onto CTFs leads to polarization enhancement, which significantly promotes charge separation and transfer (Figure 4d,e). This enhanced polarization facilitates efficient proton transfer, ultimately promoting the photosynthesis of H_2_O_2_. The triazine units within the CTFs act as the photoreduction site for the 2*e*^−^ ORR to produce H_2_O_2_, while the accumulated holes at the thiourea sites accelerate the WOR kinetics [114].

The H_2_O_2_ production rate can be further elevated to 7327 μmol·h^−1^·g^−1^ [104]. Liao et al. present the development of diazine functionalized COFs (DAzCOFs) for photosynthesis of H_2_O_2_ (Figure 5a) [104]. The study focuses on the influence of nitrogen atom positions in N-heterocycles for catalytic activity. The evaluation revealed that pyridazine embedded in TpDz exhibited a remarkable photocatalytic activity for overall H_2_O_2_ photosynthesis. The optimized H_2_O_2_ production rate of TpDz was measured to be 7327 μmol·h^−1^·g^−1^ in pure water (Figure 5b,c), ranking it as one of the best COF-based photocatalysts in photosynthesis of H_2_O_2_. Additionally, the SCC efficiency of TpDz was found to be 0.62%, surpassing natural plants. DFT calculations indicate that pyridazine acts as a superior reactive site for stabilizing endoperoxide intermediate species in the direct 2*e*^−^ ORR pathway compared to pyrimidine. Conversely, pyrazine exhibits a propensity to generate ·OOH for the two-step 1*e*^−^ ORR pathway owing to its separated nitrogen atoms.

The research pertaining to the aforementioned functional group modification primarily involves the generation of H_2_O_2_ through the ORR process. The production of H_2_O_2_ through the 2*e*^−^ WOR remains elusive in the field. However, simultaneously utilizing both ORR and WOR processes for H_2_O_2_ production would be particularly appealing, especially in terms of energy conservation and pollution reduction. Chen and co-workers incorporated acetylene and diacetylene into CTFs and successfully obtained two new CTFs, namely CTF-EDDBN and CTF-BDDBN [111]. Additionally, they demonstrate that exfoliated nanosheets derived from CTF-EDDBN and CTF-BDDEN can generate H_2_O_2_ through ORR and a new WOR process when subjected to visible light irradiation. The findings indicate that CTF-BDDBN exhibits a BET surface area of 63.5 m^2^·g^−1^, a H_2_O_2_ production rate of 97.2 μmol·h^−1^·g^−1^, and a SCC efficiency of 0.14%. DFT calculation reveals that the introduction of acetylene (−C≡C−) or diacetylene (−C≡C−C≡C−) groups into CTFs has been found to greatly enhance the photocatalytic production of H_2_O_2_. This improvement can be attributed to the presence of −C≡C− triple bonds, whose presence plays an important role in modulating the electronic structures of CTFs and suppressing charge recombination. Moreover, the incorporation of acetylene and diacetylene moieties effectively reduces the energy required for the formation of OH* and enables a new pathway for the 2*e*^−^ WOR to produce H_2_O_2_.

### 4.2. Heteroatom Doping

Heteroatom doping refers to the strategy of introducing atoms other than carbon into the framework of COFs to modify their electronic structure and enhance their photocatalytic performance. The heteroatoms, such as nitrogen, boron, or sulfur, can introduce new energy levels within the band gap of COFs, thus facilitating the transfer and separation of photogenerated charges. Moreover, the doped heteroatoms can also serve as active sites for the adsorption and activation of molecular oxygen, which is a key step in the photocatalytic production of H_2_O_2_. Additionally, heteroatom doping can also enhance structural stability, which further contributes to their photocatalytic activity. Wang et al. present the development of a fluorinated COF (TF_50_-COF) photocatalyst for the photosynthesis of H_2_O_2_ (Figure 6a) [107]. The evaluation suggests that TF_50_-COF demonstrated an AQY of 5.1% at 400 nm and a SCC efficiency of 0.17% (Figure 6g). The production of H_2_O_2_ was evaluated under different conditions. When irradiated with λ > 400 nm, light, and using ethanol as a sacrificial agent, TF_50_-COF achieved an H_2_O_2_ production rate of 1739 μmol·h^−1^·g^−1^ (Figure 6e,f). In comparison, H-COF, a non-fluorinated COF, only produced 516 μmol·h^−1^·g^−1^ of H_2_O_2_. DFT calculations were performed to comprehend the process of TF_50_-COF catalytical H_2_O_2_ production, indicating that the incorporation of F-substituents resulted in a profusion of Lewis acid sites (Figure 6d). This, in turn, affected the electronic distribution of adjacent carbon structures, resulting in the creation of highly active sites for O_2_ adsorption. Moreover, this substitution expanded the catalyst’s sensitivity to visible light, simultaneously enhancing the efficiency of charge separation.

### 4.3. Donor-Acceptor (D-A) Configuration

Enhancing the photocatalytic performance necessitates a crucial stage of fine-tuning the separation and transfer of photogenerated carriers during redox reactions. To achieve this goal, it is necessary to introduce both electron acceptors and electron donors simultaneously. These acceptors and donors are then utilized to construct an innovative framework known as donor-acceptor COFs (D-A COFs). Zhai et al. presented a study on the construction of synergistic triazine and acetylene cores in COFs for photocatalytic H_2_O_2_ production (Figure 7a) [115]. The experimental process involves the integration of triazine and acetylene units into COFs, namely EBA-COF and BTEA-COF. These COFs exhibit spatial separation of the triazine and acetylene cores, leading to efficient charge separation and suppressed charge recombination. Moreover, the C≡C linkage within the COFs facilitates the transport of electrons along the frameworks. The findings suggest that the integrated triazine and acetylene units synergistically promote H_2_O_2_ synthesis through a 2*e*^−^ ORR pathway. Specifically, the EBA-COF demonstrates an H_2_O_2_ production rate of 1830 μmol·h^−1^·g^−1^ (Figure 7d,e). Mechanism investigation demonstrated that the electron-deficient triazine moiety with a high content of nitrogen atoms exhibits a high electron affinity. Additionally, the acetylenic groups act as bridges between electron donors and acceptors, facilitating the efficient transfer of photogenerated *e*^−^ [115]. In another investigation, Cheng et al. designed covalent heptazine frameworks (CHF) with separated redox centers, aiming to enhance charge separation and achieve high H_2_O_2_ production [112]. The results revealed that the 2*e*^−^ ORR occurred on the heptazine moiety, while the 2*e*^−^ WOR took place on the acetylene or diacetylene bond in the CHF. The exciton binding energy of the diacetylene-containing polymer was found to be 24 meV. Under simulated solar irradiation, the rationally designed CHFs achieved a SCC efficiency of 0.78%. Additionally, mechanism investigation showed that the significant differences in electronegativity between the s-heptazine moiety and different biphenyl compounds facilitated efficient charge transfer from peripheral groups to the central s-heptazine moiety [112].

## 5. Outlook and Perspective

In the last three years, COFs have emerged as promising photocatalysts for the synthesis of H_2_O_2_ due to their intrinsic properties such as tunable porosity, structural diversity, and modifiable electronic structure. The exploration of COFs in this realm has unveiled a spectrum of results, demonstrating enhanced photocatalytic activity and stability under various operational conditions. Notably, the strategic incorporation of active sites and the optimization of electronic properties are important in amplifying the photocatalytic efficiency of COFs for H_2_O_2_ production. Currently, reported active sites for the ORR include pyridine, bipyridine, pyridazine, benzene, diarylamine, pyrene, etc., while active sites for the WOR include diacetylene, triazine, acetylene, bipyridine, and others.

Looking forward, the prospects of COFs in photocatalytic H_2_O_2_ synthesis appear to be expansive yet challenging. Some challenges still exist in this field. The pore size of most reported COFs for photocatalytic H_2_O_2_ production falls within the 1 to 3 nm range. However, there is still a relative lack of research on how the size of these pores affects the photocatalytic process. Additionally, the stability of COFs also restricts their industrial applications. COFs can sometimes suffer from lower stability, especially in aqueous environments, which may limit their practical applications. Compared to inorganic photocatalysts, COFs may not match the robustness and long-term stability of these more established materials. To enhance their stability, more appropriate building blocks and linkage chemistry can be selected or fabricate composites by combining COFs with materials possessing better stabilities. Future research should delve deeper into the mechanistic pathways of photocatalytic reactions of COFs, which is important for rational design and functionalization strategies. Moreover, addressing the scalability and economic viability of COF-based photocatalytic systems is imperative to transition from laboratory-scale research to real-world applications. The exploration of novel COF structures, the incorporation of alternative active sites, and the development of hybrid systems that combine the advantages of various photocatalytic materials will potentially pave the way for breakthroughs in sustainable H_2_O_2_ production.

## Figures and Tables

**Figure 1 polymers-16-00659-f001:**
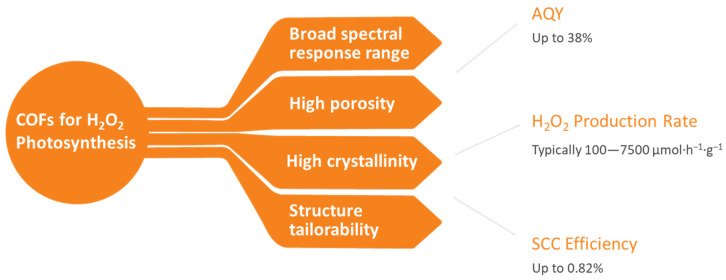
The advantages of COFs for the photosynthesis of H_2_O_2_.

**Figure 2 polymers-16-00659-f002:**
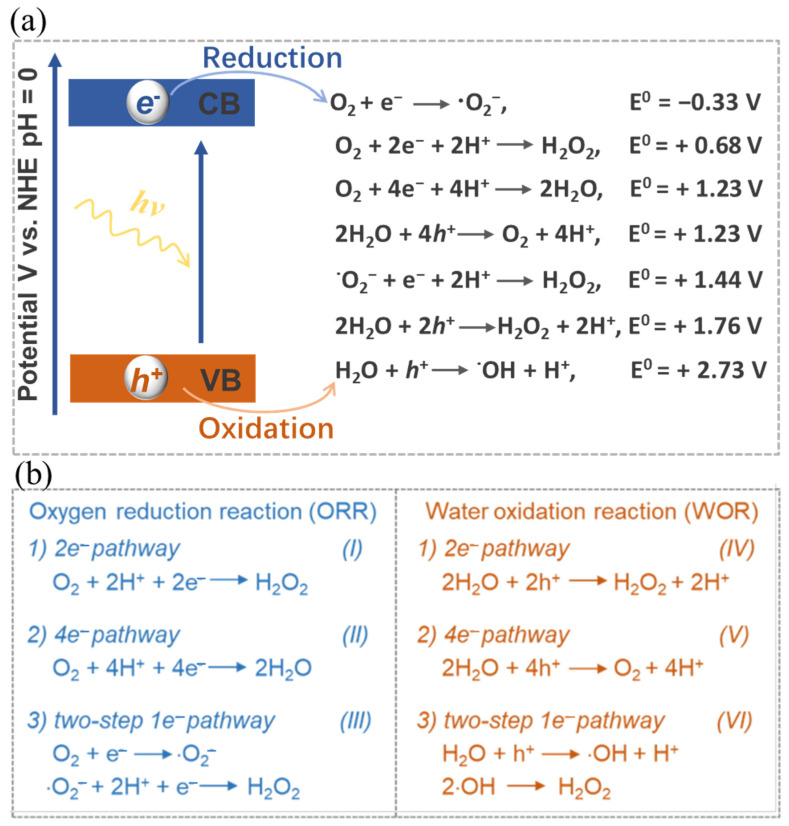
(**a**) Schematic illustration and (**b**) Corresponding energy diagrams of the oxygen reduction and water oxidation involved in H_2_O_2_ photosynthesis. Reproduced with permission [1].

**Figure 3 polymers-16-00659-f003:**
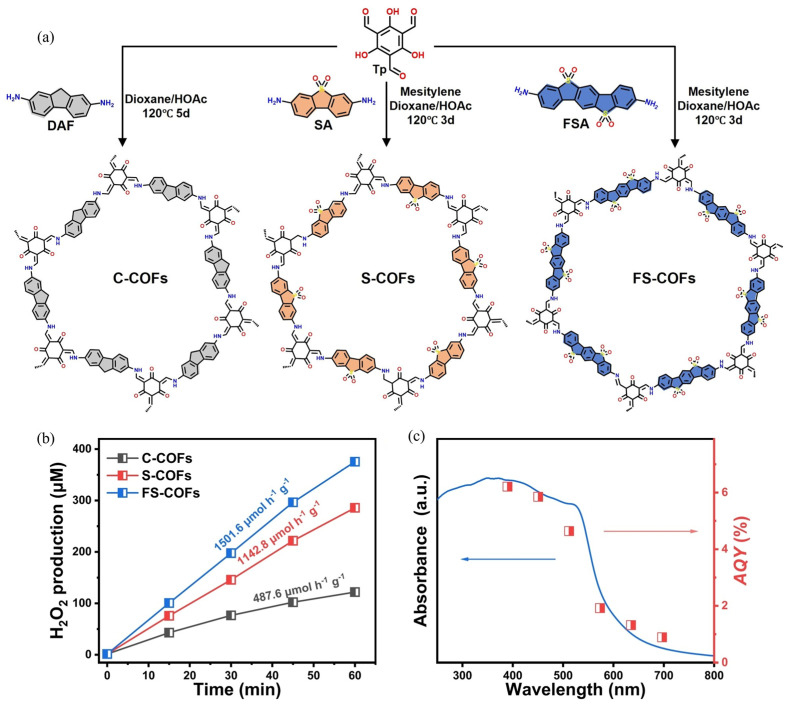
(**a**) Synthetic process of sulfone-modified COFs. (**b**) Photocatalytic H_2_O_2_ production rate of C-COFs (black), S-COFs (red), and FS-COFs (blue). (**c**) UV/Vis spectrum of FS-COFs and AQY for H_2_O_2_ production. Reproduced with permission [113]. Copyright 2023, Wiley-VCH.

**Figure 4 polymers-16-00659-f004:**
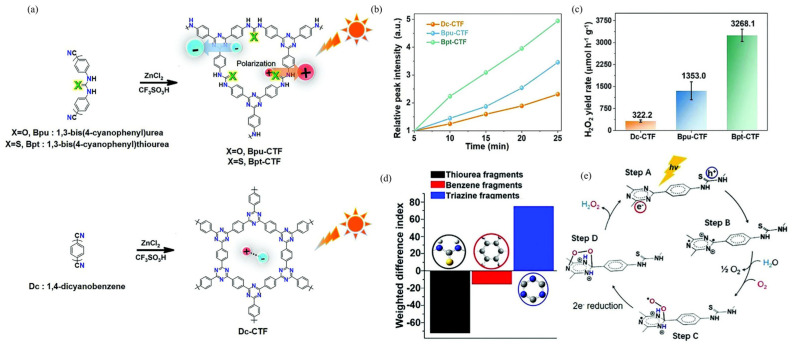
(**a**) Synthetic route of thiourea-functionalized COFs. (**b**) Relative OH bending intensity of thiourea-functionalized COFs. (**c**) H_2_O_2_ generation rate. (**d**) Weight differential index of fragments in excited states 4–9 of Bpt-CTF. (**e**) Schematic diagram of the catalytic pathway. Reproduced with permission [114]. Copyright 2022, Wiley-VCH.

**Figure 5 polymers-16-00659-f005:**
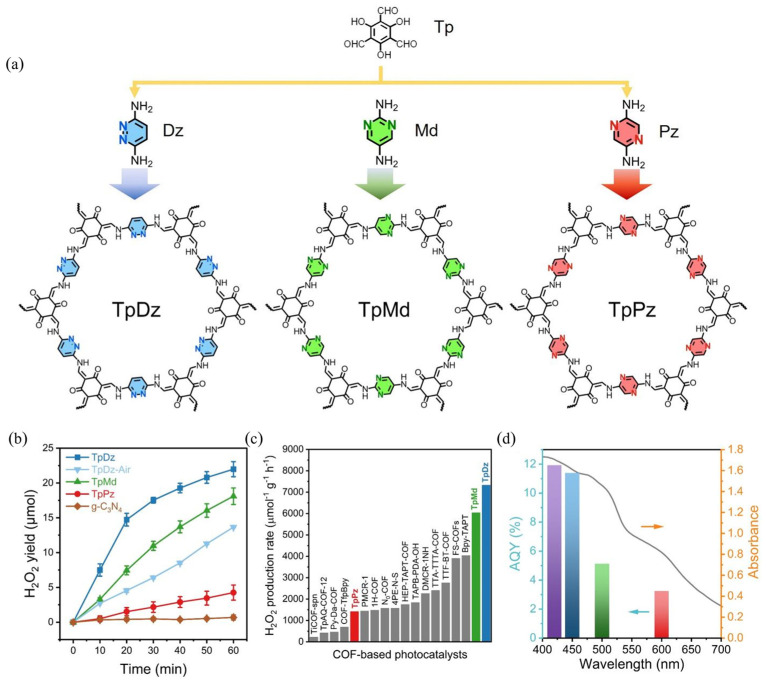
(**a**) Schematic diagram of the synthesis process. (**b**) H_2_O_2_ production yield. (**c**) H_2_O_2_ production rate. (**d**) AQY of TpDz at chosen wavelengths: 420 (purple), 450 (light blue), 500 (green), and 600 (red) nm. Reproduced with permission. Copyright 2023 [104], Wiley-VCH.

**Figure 6 polymers-16-00659-f006:**
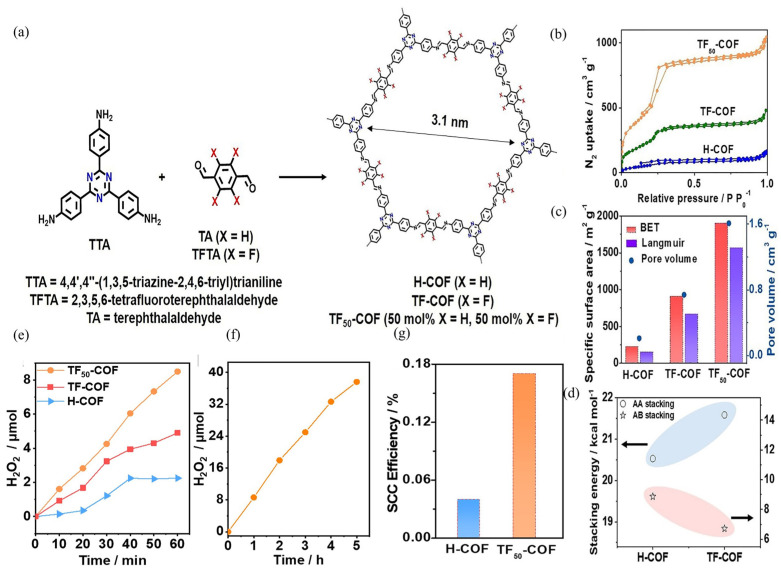
(**a**) Synthetic process of fluorinated COFs. (**b**) N2 sorption isotherms of obtained COFs. (**c**) Specific surface areas and pore volumes in H-COF, TF-COF, and TF50-COF Using BET and Langmuir Models. (**d**) Crystal stacking energies of TF-COF and H-COF. (**e**) Photocatalytic H_2_O_2_ production rate for 1 h. (**f**) Photocatalytic H_2_O_2_ production rate for 5 h. (**g**) The SCC efficiencies of synthesized COFs. Reproduced with permission [107]. Copyright 2022, Wiley-VCH.

**Figure 7 polymers-16-00659-f007:**
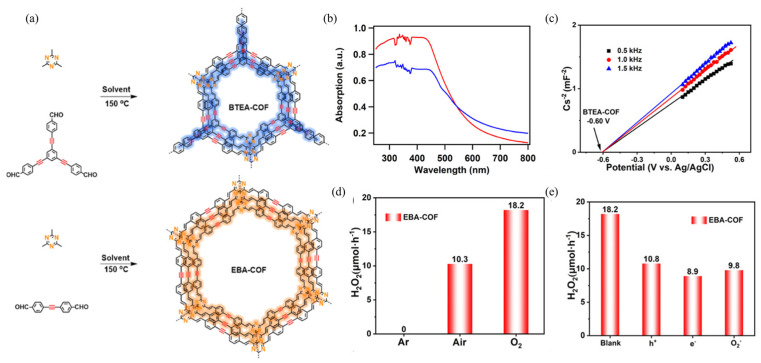
(**a**) Synthetic route of BTEA-COF (red) and EBA-COF (blue). (**b**) UV-DRS of EBA-COF (blue) and BTEA-COF (red). (**c**) Mott-Schottky plot of the BTEA-COF. (**d**) Reactions of EBA-COF under varied gas atmospheres. (**e**) Photocatalytic production of H_2_O_2_ using EBA-COF in the presence of KI, CuSO_4_, and p-benzoquinone. Reproduced with permission [115]. Copyright 2022, American Chemical Society.

**Table 1 polymers-16-00659-t001:** Representative COF photocatalysts for H_2_O_2_ generation (SCC, AQY, EtOH, BA, and MeOH are abbreviations for solar-to-chemical energy conversion, apparent quantum yield, ethanol, benzyl alcohol, and methanol, respectively; “/” means the data is not provided in the literature).

Entry	Light Source (nm)	H_2_O_2_ Generation Rate (μmol·h^−1^·g^−1^)	SCC Efficiency (%)	AQY (%) at 420 nm	Reaction Pathway	Solvent	Ref.
TAPD-omeCOF	420–700	97 ± 10	/	/	2*e*^−^ ORR	H_2_O:EtOH (9:1)	[20]
TAPD-meCOF	420–700	97 ± 10	/	/	2*e*^−^ ORR	H_2_O:EtOH (9:1)	[20]
TF_50_-COF	λ > 400	1739	0.17	5.1	Two-step 1*e*^−^ ORR	H_2_O:EtOH (9:1)	[21]
TTF-BT-COF	420–700	2760	0.49	11.19	2*e*^−^ ORR 2*e*^−^ WOR	H_2_O	[22]
EBA-COF	λ = 420	1820 2550	/	4.4	Two-step 1*e*^−^ ORR	H_2_O:EtOH (9:1) H_2_O:BA (9:1)	[23]
HEP-TAPT-COF	λ > 420	1750	0.65	15.35	2*e*^−^ ORR	H_2_O	[24]
Cof-TpDz	λ > 420	7327	0.62	11.9	2*e*^−^ ORR	H_2_O	[25]
Py-Da-COF	420–700	461 682 1242		/ 2.4 4.5	2*e*^−^ ORR	H_2_O H_2_O:EtOH (9:1) H_2_O:BA (9:1)	[14]
FS-COF	λ > 420	3904.2	/	6.21	2*e*^−^ ORR	H_2_O	[26]
TZ-COF	λ > 420	268 386 4951	0.036	0.6	2*e*^−^ ORR	H_2_O H_2_O:MeOH (1:1) H_2_O:BA (1:1)	[27]
TAPT-TFPA COFs	Xe lamp	2143	0.82	6.5 at 400 nm	Two-step 1*e*^−^ ORR	H_2_O:EtOH (9:1)	[28]
CHF-DPDA	λ > 420	1725	0.78	16	2*e*^−^ ORR 2*e*^−^ WOR	H_2_O	[29]
CTF-BDDBN	λ > 420	97.2	0.14	/	2*e*^−^ ORR 2*e*^−^ WOR	H_2_O:MeOH (9:1)	[30]
COF-TfpBpy	λ > 420	695	0.57	8.1	2*e*^−^ ORR 2*e*^−^ WOR	H_2_O	[31]
COF-N32	λ > 420	605	0.31	6.2 at 459 nm	2*e*^−^ ORR 2*e*^−^ WOR	H_2_O	[32]
Bpt-CTF	350–780	3268.1	/	6.4	2*e*^−^ ORR	H_2_O	[33]
sonoCOF-F2	λ > 420	2736	/	4.8	Two-step 1*e*^−^ ORR	H_2_O	[34]

## Data Availability

Data are contained within the article.

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
