# Peer review of "COF-Based Photocatalysts for Enhanced Synthesis of Hydrogen Peroxide"

_polymers, 2024, doi:10.3390/polym16050659_

Round 1
Reviewer 1 Report
Comments and Suggestions for Authors
This manuscript by Prof. Tan et al. reviews the recent progress for the synthesis of H2O2 by covalent organic frameworks (COFs) under the photolysis condition. Many industrial processes have been developed to the preparation of H2O2 while substantial energy consumption, the use of hazardous agents and the generation of waste prompt the development of sustainable methods.
In this review article, the author outlines the fundamentals for H2O2 production by photocatalysis. Here every possible pathway is carefully addressed, including bottlenecks in the pathways. Followed is the discussion of COF materials used for photocatalytic H2O2 production. Strategies by structural designs are discussed for the enhancement of H2O2 production. This review article provides comprehensive information on the photocatalytic production of H2O2 by COF materials. The review serves as a starting point for those who would like to work on the specific topic. However, currently, it is difficult for a starter to obtain enough information from this review. It misses some important aspects. The following questions have to be addressed before it can be considered for publication.
This Reviewer has a few suggestions to be included in the manuscript.
(1) When dealing with the electron transfer from the photoexcited COF units toward H2O2 production, the energy diagram of the frontier MOs of COFs, O2 and H2O should be depicted.
(2) In the mechanism, there are several possible intermediate species. It may be interesting to discuss them. In addition, the method and reagents that are used to clarify the involvement of any intermediate species can be mentioned.
(3) Stability of the COF materials should be addressed if they are to be used in industrial applications. Any way to improve the stability of the materials?
(4) What are the main categories of COFs? What are the typical synthetic methods for COFs? The authors need to put this part in the review.
(5) How about the stability of COFs photocatalysts and the comparison to other photocatalysts? The authors should include this discussion.
(6) How does pore size affect photocatalytic activity?
(7) What are the typical methods to test H2O2 for the COFs studies?
(8) The electronic structures are not really well discussed. But you mentioned in the Abstract that “We mainly delve into the structural modulation of COFs, elucidating the correlation between their structures and electronic properties with photocatalytic performance.”
(9) In Fig. 1, how do you get the AQY to be 38%?? This is not mentioned.
(10) Some figures have poor layout, please polish.
(11) English needs polish.
There are a few minor issues requiring the author’s attention.
(1) Line 263, “nom” --> “non”
(2) For all figures, outer lines around the images should be removed.
Comments on the Quality of English Language
It needs checking and polishing
Author Response
We thank you for the helpful comments and suggestions, and all the issues raised were essentially accepted. According to your comments, we have addressed all the concerns. The changes in the manuscript have been highlighted in yellow for your convenience. In the following point-by-point response, we endeavor to address all the questions and hope that the revision is satisfactory.

Reviewer 2 Report
Comments and Suggestions for Authors
This manuscript by Prof. Tan et al. reviews the recent progress for the synthesis of H2O2 by covalent organic frameworks (COFs) under the photolysis condition. Many industrial processes have been developed to the preparation of H2O2 while substantial energy consumption, the use of hazardous agents and the generation of waste prompt the development of sustainable methods.
In this review article, the author outlines the fundamentals for H2O2 production by photocatalysis. Here every possible pathway is carefully addressed, including bottlenecks in the pathways. Followed is the discussion of COF materials used for photocatalytic H2O2 production. Strategies by structural designs are discussed for the enhancement of H2O2 production. This review article provides comprehensive information on the photocatalytic production of H2O2 by COF materials. The review serves as a starting point for those who would like to work on the specific topic. This Reviewer recommends the acceptance of the manuscript by the Journal.
This Reviewer has a few suggestions to be included in the manuscript.
(1) When dealing with the electron transfer from the photoexcited COF units toward H2O2 production, the energy diagram of the frontier MOs of COFs, O2 and H2O should be depicted.
(2) In the mechanism, there are several possible intermediate species. It may be interesting to discuss them. In addition, the method and reagents that are used to clarify the involvement of any intermediate species can be mentioned.
(3) Stability of the COF materials should be addressed if they are to be used in industrial applications. Any way to improve the stability of the materials?
There are a few minor issues requiring the author’s attention.
(1) Line 263, “nom” --> “non”
(2) For all figures, outer lines around the images should be removed.
Author Response

(The authors gave the same response as above.)
